# FDG-PET in Chimeric Antigen Receptor T-Cell (CAR T-Cell) Therapy Toxicity: A Systematic Review

**DOI:** 10.3390/cancers16091728

**Published:** 2024-04-29

**Authors:** Akram Al-Ibraheem, Ahmed Saad Abdlkadir, Egesta Lopci, Sudqi Allouzi, Diana Paez, Maryam Alkuwari, Mohammad Makoseh, Fuad Novruzov, Sharjeel Usmani, Kamal Al-Rabi, Asem Mansour

**Affiliations:** 1Department of Nuclear Medicine and PET/CT, King Hussein Cancer Center (KHCC), Al-Jubeiha, Amman 11941, Jordan; ahmedshukri92@hotmail.com (A.S.A.); sa.16577@khcc.jo (S.A.); 2School of Medicine, University of Jordan, Amman 11942, Jordan; 3Nuclear Medicine Unit, IRCCS—Humanitas Research Hospital, Via Manzoni 56, 20089 Rozzano, Italy; egesta.lopci@gmail.com; 4Nuclear Medicine and Diagnostic Imaging Section, Division of Human Health, Department of Nuclear Sciences and Applications, International Atomic Energy Agency, 1220 Vienna, Austria; d.paez@iaea.org; 5Clinical Imaging Department, Hamad Medical Corporation, Doha 7GPR+3M9, Qatar; malkuwari5@hamad.qa; 6Department of Medical Oncology, King Hussein Cancer Center (KHCC), Amman 11941, Jordan; mm.09744@khcc.jo (M.M.); ka.10798@khcc.jo (K.A.-R.); 7Department of Nuclear Medicine, The National Centre of Oncology, Ministry of Health of Azerbaijan Republic, Baku AZ1012, Azerbaijan; fuad.novruzov@ntm.az; 8Sultan Qaboos Comprehensive Cancer Care and Research Centre, Muscat 5661, Oman; dr_shajji@yahoo.com; 9Department of Radiology, King Hussein Cancer Center (KHCC), Al-Jubeiha, Amman 11941, Jordan; amansour@khcc.jo

**Keywords:** immune-related adverse events, CAR T cells, chimeric antigen receptor, cytokine release syndrome, immune effector cell-associated neurotoxicity, ICANS, CRS, PET, positron emission tomography

## Abstract

**Simple Summary:**

This manuscript systematically reviews the role of positron emission tomography (PET) in the assessment of toxicity associated with chimeric antigen receptor T-cell therapy (CAR T-cell therapy). CAR T-cell therapy, a revolutionary form of immunotherapy, activates immune mechanisms against malignant cells. By doing so, it can also provoke immune-mediated responses against healthy tissues. These responses exhibit diverse clinical, radiological, and functional manifestations across multiple physiological systems. Adverse events include cytokine release syndrome and immune effector cell-associated neurotoxicity syndrome. Both can cause life-threatening complications. Given the critical imperative of timely identification and vigilant monitoring of these adverse events to enable targeted interventions, PET has emerged as an indispensable imaging biomarker facilitating their detection, prediction, and surveillance. Hence, our investigation is specifically tailored to examine the utility of PET in evaluating adverse events induced by CAR T-cell therapy.

**Abstract:**

The utilization of chimeric antigen receptor (CAR) T-cell therapy to target cluster of differentiation (CD)19 in cancer immunotherapy has been a recent and significant advancement. Although this approach is highly specific and selective, it is not without complications. Therefore, a systematic review was conducted to assess the current state of positron emission tomography (PET) in evaluating the adverse effects induced by CAR T-cell therapy. A thorough search of relevant articles was performed in databases such as PubMed, Scopus, and Web of Science up until March 2024. Two reviewers independently selected articles and extracted data, which was then organized and categorized using Microsoft Excel. The risk of bias and methodological quality was assessed. In total, 18 articles were examined, involving a total of 753 patients, in this study. A wide range of utilities were analyzed, including predictive, correlative, and diagnostic utilities. While positive outcomes were observed in all the mentioned areas, quantitative analysis of the included studies was hindered by their heterogeneity and use of varying PET-derived parameters. This study offers a pioneering exploration of this promising field, with the goal of encouraging further and more focused research in upcoming clinical trials.

## 1. Introduction

The emergence of cancer immunotherapies signifies a promising therapeutic strategy in the field of oncology. Unlike traditional cancer treatments, these therapies exert direct tumoricidal effects instead of targeting lymphocyte receptors or their ligands [1]. Recent advancements in cancer immunotherapy, including the approval of chimeric antigen receptor (CAR) T-cell therapy, have significantly progressed in the field [2]. CAR T-cell therapy involves modifying a patient’s own T cells to express synthetic receptors that target specific tumor antigens, which are then reintroduced into the patient following lymphocyte depletion through chemotherapy. This approach combines the benefits of monoclonal antibody therapy and cytotoxic T cells to initiate a targeted immune response against tumor cells [3].

CAR T-cell therapy involves a multi-step process starting with the collection of T cells, which are critical components of the immune system, from the patient’s blood. These T cells are then transported to a specialized laboratory where they undergo genetic modification [4]. During this phase, a gene that encodes for a specific CAR is introduced into the T cells. This CAR is a synthetic receptor designed to bind to a particular antigen on the tumor cells. The genetic modification is typically achieved using viral vectors, which are efficient at delivering the CAR gene into the T cells’ DNA. Once the T cells are engineered to express the CAR on their surface, they are known as CAR T cells. These cells are then cultivated in the lab to increase their numbers significantly, a process known as expansion. Following expansion, the CAR T cells are infused back into the patient’s bloodstream. Upon reinfusion, the engineered CAR T cells navigate through the body, identifying and attaching to cancer cells that exhibit the target antigen on their surfaces. The binding of the CAR T cells to the cancer cells triggers a potent immune response, leading to the destruction of the targeted cancer cells [4]. This method not only provides a direct attack on cancer cells but also offers the potential for long-lasting immunological memory, which can continue to protect against cancer recurrence. The specificity and efficacy of CAR T-cell therapy make it a promising treatment option, particularly for patients who have not responded to conventional therapies (Figure 1).

Currently, the Food and Drug Administration (FDA), and the European Medicines Agency (EMA) have granted approval for several CAR T-cell therapies that target the cluster of differentiation (CD)19 antigen on B cells. In 2017, two CAR T-cell agents, namely, axicabtagene ciloleucel and tisagenlecleucel, were approved [5,6]. They were both granted EMA approval in the following year. Shortly thereafter, brexucabtagene autoleucel received approval from the EMA in 2019, followed by FDA approval in 2020 [7]. Lastly, lisocabtagene maraleucel received EMA and FDA approvals in 2019 and 2021, respectively [8]. Tisagenlecleucel, axicabtagene ciloleucel, and lisocabtagene maraleucel have been approved for adult patients with diffuse large B-cell lymphoma (DLBCL) who have experienced relapse or have not responded to two or more lines of systemic therapy [9]. Tisagenlecleucel has also been approved for the treatment of patients under the age of 25 with relapsed or refractory (r/r) B-cell precursor acute lymphoblastic leukemia (B-ALL) [9]. Additionally, brexucabtagene autoleucel can be utilized for the treatment of adult patients with relapsed or refractory mantle cell lymphoma [9]. The FDA and EMA have granted approval to idecabtagene vicleucel and ciltacabtagene autoleucel in 2021 and 2022, respectively, for the treatment of r/r multiple myeloma [10,11].

The utilization of 2-deoxy-2-[^18^F]fluoro-D-glucose positron emission tomography/computed tomography ([^18^F]FDG PET/CT) in molecular imaging is essential for diagnosing, staging, assessing therapeutic response, and evaluating recurrence in patients with metabolically active lymphoma [12,13]. The available literature with regards to CAR T-cell therapy provides information on the application of [^18^F]FDG PET/CT in CAR T-cell therapy at the initial stage, with two scans conducted prior to the initiation of treatment, specifically emphasizing the time of decision (TD) and time of transfusion (TT), respectively [14]. Subsequently, two supplementary [^18^F]FDG PET/CT scans are utilized for the assessment of the therapy response [15]. The initial [^18^F]FDG PET/CT scan is performed after one month (M1) of therapy, followed by another scan at three months (M3). These scans exhibit a sensitivity rate of 99% and a specificity rate of 100% when initially evaluating the response to treatment [16,17]. Therefore, [^18^F]FDG PET/CT is a reliable imaging modality for the monitoring and follow-up of patients receiving CAR T-cell therapy, with increasing evidence supporting its reliability [18].

The potential for the growth and development of CAR T-cell therapy is expected to increase with the advancement of new treatments and enhancements in the management of adverse effects. Currently, there is a lack of recognition of the possible utility of [^18^F]FDG PET/CT in assessing the adverse effects of CAR T-cell therapy. Thus, our objective is to systematically review the use of PET imaging in the detection and evaluation of toxicities resulting from CAR T-cell therapy.

## 2. Materials and Methods

### 2.1. Systematic Literature Search

In this systematic review, we conducted a thorough examination of relevant studies following the guidelines outlined in the Preferred Reporting Items for Systematic Review and Meta-Analysis Protocols (PRISMA-P) statement [19]. Two authors, namely, A.A.-I. and A.S.A., independently conducted electronic searches of the PubMed (Bethesda, Maryland, United States), Scopus (Amsterdam, Netherlands), and Web of Science (Philadelphia, Pennsylvania, United States) databases to identify published manuscripts that explored the role of PET in detecting CAR T-cell therapy-induced toxicity. Our search strategy included various terms related to PET, CAR T-cell therapy, and adverse events and was last updated on 22 March 2024. Several methods were used to overcome technique limitations. Initial electronic searches focused on PubMed (Bethesda, Maryland, United States), Scopus (Amsterdam, Netherlands), and Web of Science (Philadelphia, Pennsylvania, United States). Additional databases and grey literature sources were included to broaden the search and reduce publication bias. To ensure data harvest and analysis linguistic uniformity, non-English research was removed during the first screening.

Only articles that specifically addressed the application of PET in the context of CAR T-cell therapy toxicity in real clinical settings were included in our analysis. During the initial screening process, any duplicate studies, book chapters, conference papers, abstracts, preclinical studies, or irrelevant articles were excluded. The full texts of potentially relevant studies were retrieved for detailed examination. A cross-reference search was conducted to ensure that all relevant studies were included. The retrieved data were loaded into Microsoft Excel Professional Plus 2021 software (Redmond, Washington, United States). The software was utilized to organize, sort, screen, and filter the retrieved data.

### 2.2. Assessment of Methodological Quality

Next, the methodological quality of the included articles discussing diagnostic utility was independently assessed using the standardized Quality Assessment of Diagnostic Accuracy Studies (QUADAS-2) protocol for studies exploring diagnostic utilities [20]. The QUADAS-2 scores were assessed to determine the risk of bias and methodological applicability. These scores were then documented and organized in a table for all the studies included in the analysis. Other articles exploring predictive capabilities were evaluated using the Quality in Prognostic Studies (QUIPS) tool [21]. For single reported cases, we implemented the CAse Report (CARE) guideline criteria to assess the reliability of the information presented [22]. A set of 13 criteria outlined in the CARE checklist were utilized for this evaluation [23]. In the interest of quality control, a system of quality grading was established, wherein the quality of a paper is appraised according to the degree of fulfillment of the specified criteria. To assess the quality of the reported cases, the two authors adapted a CARE quality scoring system [24]. High-quality papers should fulfill a minimum of 10 criteria, while low-quality papers should address no more than 4. Papers of intermediate quality should meet a range of 5–9 criteria.

For each article, information such as the year of publication, first author, country of origin, study design, objectives, study population characteristics, adopted radiotracer, and main findings were extracted.

Finally, although two authors conducted the searches and quality ratings individually, they resolved any discrepancies through conversation and agreement. This method improves review reliability. These approaches improved the systematic review methodological rigor and credibility.

## 3. Results

### 3.1. Search Results

This systematic review initially identified a total of 127 articles from three different databases (PubMed: 45 articles; Scopus: 40 articles; and Web of Science: 42 articles). After removing 35 duplicates, 92 titles and abstracts were screened, with the majority of the articles not aligning with the study’s objectives, primarily focusing on CAR T-cell therapeutic efficacy. Ultimately, only 17 articles met the criteria for inclusion and were further evaluated after retrieving the full-text versions [25,26,27,28,29,30,31,32,33,34,35,36,37,38,39,40,41]. An additional study [42] was found during the cross-reference check, bringing the total number of qualified studies to 18 for further analysis (Figure 2A).

### 3.2. Assessment of Methodological Quality and Risk of Bias

A total of 11 studies were identified exploring the predictive capabilities of PET in CAR T-cell toxicity. Utilizing the QUIPS tool, two studies were found to exhibit a moderate risk of bias in the study attrition domain [25,31], and two were discovered to have a moderate risk of bias in the study confounding domain [28,31]. Four other studies [26,36,39,41] had a high risk of bias in the study confounding domains (Figure 2B).

With regards to the QUADAS-2 criteria, only four studies were found to be eligible for evaluation, as they implemented PET in diagnosing CAR T-cell therapy toxicity in a subset of patients [29,30,32,40]. One study exhibited a high risk of bias in patient selection [40], while another revealed a similar bias risk in relation to the index test employed (Figure 2C) [32]. Furthermore, assessments of applicability concern consistently yielded results indicating a low risk, save for occasional instances of elevated concern. Notably, a single study raised high-risk apprehensions pertaining to patient selection [29], while another study elicited similar concerns regarding the index test (Figure 2D) [32].

The three remaining articles detailed the utilization of PET in a singular context, specifically for assessing the toxicity associated with CAR T-cell therapy. This subset comprised a clinical case presentation [33], a comprehensive review article [42], and an original prospective research study [34]. As previously mentioned, assessments of the articles’ reporting quality were performed utilizing the CARE criteria. In each instance, the evaluated articles were deemed to possess an intermediate level of reporting quality (Table 1).

## 4. Systematic Review

### 4.1. Study Characteristics

A total of 18 studies encompassing 753 patients were systematically reviewed. Of these, 16 studies were original research [25,26,27,28,29,30,31,32,34,35,36,37,38,39,40,41]. The predominant method used in the studies was a retrospective analysis [25,26,27,28,31,32,35,36,37,38,39,40,41], with only three studies utilizing a prospective approach [29,30,34]. All the studies focused on assessing individuals with hematologic malignancies, particularly non-Hodgkin’s lymphoma (NHL). The majority of the studies were conducted in Europe, with single-center studies being more common than multicenter collaborations. The main objective of most of the studies was to investigate the predictive ability of PET in detecting toxicity related to CAR T-cell therapy, with some also examining its diagnostic value. It is worth noting that all the studies utilized [^18^F]FDG PET/CT as the primary adopted radiotracer (Table 2).

### 4.2. Baseline PET for the Prediction of CAR T-Cell Therapy Toxicity

Numerous studies have explored the utility of volumetric PET-derived metrics in predicting CAR T-cell therapy toxicity, focusing on metabolic tumor volume (MTV) and total lesion glycolysis (TLG).

#### 4.2.1. First Clinical Study

In 2019, Wang et al. initiated a retrospective investigation, primarily assessing the potential of baseline [^18^F]FDG PET/CT to predict CAR T-cell therapy toxicity. Among 19 patients with NHL, 42.1% experienced grade 0 or 1 cytokine release syndrome (CRS), 36.8% experienced grade 2, and 21.1% experienced grade 3 CRS. A [^18^F]FDG PET/CT analyses revealed significantly higher baseline MTV and TLG values in patients with severe CRS compared to non-severe cases. Furthermore, neurological toxicity, the second most common adverse event, affected 21.1% of the patients, with no significant difference in the MTV or TLG values between those with and without neurological toxicity. These findings underscore the potential of [^18^F]FDG PET/CT-derived parameters in predicting and stratifying CAR T-cell therapy-associated toxicities, particularly CRS severity, aiding in treatment management and patient care optimization [31]. In contrast to the preceding research, Derlin et al. conducted a retrospective study involving ten patients diagnosed with NHL, utilizing serial [^18^F]FDG PET/CTs [41]. This study focused on analyzing the metabolic parameters of lymphoma and lymphoid organs in patients receiving Tisagenlecleucel for r/r DLBCL. Among these patients, 40% developed neurotoxicity. Interestingly, neither the MTV nor TLG values demonstrated significant elevation in the patients experiencing neurotoxicity (*p* = 0.148 and *p* = 0.101, respectively). However, the highest value of baseline maximum standardized uptake value (SUVmax), a semi-quantitative PET metric, exhibited a significant increase in the patients with neurotoxicity (*p* = 0.048) [41]. Nevertheless, it is crucial to acknowledge the limitations of this study, including its retrospective design, monocentricity, and restricted statistical robustness. These constraints necessitate cautious interpretation of the findings and highlight the need for further comprehensive investigations in this domain.

#### 4.2.2. Unicentric Studies with Endorsing Evidence

Approximately two years later, Hong et al. conducted a retrospective analysis involving 41 patients with r/r NHL undergoing CAR T-cell therapy, yielding insightful findings [26]. Utilizing [^18^F]FDG PET/CT, they measured the baseline mean standardized uptake value (SUVmean), MTV, and TLG of lymphomatous lesions. Among these patients, 82.9% experienced CRS, with 14.6% encountering grade 3 CRS. However, no grade 4–5 CRS cases were observed. The patients were stratified based on the optimal cutoff values of their PET metabolic parameters. Univariate and multivariate analyses revealed the baseline SUV as an independent risk factor for CRS (*p* = 0.034). Moreover, the patients with elevated baseline PET metrics, including MTV, TLG, and SUVmean, exhibited a higher incidence of severe CRS. Notably, higher baseline PET metabolic parameters correlated with an increased risk of coagulation disorders, evidenced by a prolonged prothrombin time and activated partial prothrombin time in the groups with a baseline MTV value ≥ 26.37 cm^3^, SUVmean ≥ 4.36, and TLG ≥ 78.61 [26]. These findings underscore the potential of PET metabolic parameters in predicting severe CRS and associated coagulation abnormalities following CAR T-cell therapy in r/r NHL patients. 

Morbelli et al. retrospectively studied 21 refractory DLBCL patients who underwent whole-body and brain [^18^F]FDG PET scans before and after CAR T-cell therapy [28]. Among the patients, 11 developed CRS and 5 developed immune effector cell-associated neurotoxicity syndrome (ICANS). The patients who developed ICANS had significantly higher baseline MTV and TLG values compared to those who developed CRS (*p* < 0.02) [28]. Ababneh et al. conducted a retrospective study in a single center to assess the predictive value of metabolic parameters measured using [^18^F]FDG PET on toxicities related to CAR T-cell therapy [39]. The study included 59 patients with r/r large B-cell lymphoma (LBCL) and found that PET parameters were able to correlate with adverse events associated with CAR T-cell therapy. Specifically, a higher baseline TLG was found to be associated with CRS (*p* = 0.04). On the other hand, a higher baseline MTV was correlated with the development of ICANS, and a higher baseline SUVmax prior to CAR T-cell therapy was associated with grade 3–4 neurological events (*p* = 0.01 for each) [39]. Recently, Ligero et al. introduced a PET-based radiomic signature using machine learning techniques [37]. This single-center study included 93 consecutive patients with r/r LBCL who underwent a CAR T-cell infusion between 2018 and 2021. The cohort was divided into a training set (73 patients) and a test set (20 patients). The radiomic features were extracted from the baseline PET scans of the patients. Despite demonstrating superiority over conventional PET-derived metrics such as the highest SUVmax and MTV values, this radiomic signature model did not achieve statistical significance in predicting CAR T-cell therapy toxicity [37].

#### 4.2.3. Unicentric Studies with Negative Results

A multidisciplinary Spanish group of physicians led by Iacoboni et al. [36] conducted a monocentric retrospective analysis involving 35 patients with r/r LBCL. They assessed the total MTV and highest SUVmax at baseline for their predictive capabilities. The results revealed that the baseline total MTV and highest SUVmax were not significantly linked with grade ≥ 2 CRS, ICANS, or other clinically significant CAR T-cell therapy toxicities. Specifically, the incidence of grade ≥ 2 toxicity was 33.3% in the patients with a high baseline total MTV compared to 12.5% in those with a low total MTV (*p* = 0.39). Regarding SUVmax, the adverse event rates were 33.3% in patients with low (<20) and 26.1% in those with high (≥ 20) values (*p* = 0.71) [36]. These findings suggest that while baseline total MTV and highest SUVmax did not significantly predict severe toxicities, further investigation is warranted to delineate their potential role in predicting adverse events following CAR T-cell therapy in r/r LBCL patients.

Similarly, Cohen et al. found no statistically significant association between PET parameters and the occurrence of CRS or ICANS [35]. The authors conducted a retrospective analysis focusing on metabolic PET parameters including baseline SUVmax, total MTV, and TLG in 48 patients with r/r DLBCL. Post-therapy, incidences of any-grade CRS, grades 3–4 CRS, and ICANS were reported in 76.2%, 11.9%, and 21.4% of patients, respectively [35].

#### 4.2.4. Multicentric Studies

Voltin et al. conducted a German multicentric retrospective study involving 47 patients with LBCL to explore the correlation between total MTV and CAR T-cell therapy toxicity [38]. Their analysis did not reveal any significant association between elevated metabolic tumor burden and the occurrence of CRS or ICANS [38].

Another multicentric retrospective study conducted in France by Marchal et al. [27] yielded significant results. The study included 56 patients with LBCL who were treated with tisagenlecleucel and axicabtagene ciloleucel. Researchers have found a significant correlation between baseline SUVmean and the severity of CAR T-cell therapy toxicity. Specifically, a baseline liver SUVmean > 2.5 was linked to grades 2–4 CRS, while a baseline spleen SUVmean > 1.9 was associated with grades 2–4 ICANS [27].

#### 4.2.5. Latest Research Evidence

In the most recent retrospective study by Gui et al., 38 patients diagnosed with DLBCL who underwent CAR T-cell therapy were included [25]. These patients had undergone baseline [^18^F]FDG PET/CT within three months prior to CAR T-cell therapy infusion, during which highest SUVmax, TLG, and MTV values were recorded. Spearman’s rank correlation analysis indicated a strong correlation between CRS grade and pre-infusion SUVmax (Spearman’s rho = 0.806, *p* < 0.001) and a moderate correlation with pre-infusion TLG (Spearman’s rho = 0.534, *p* < 0.001). Multinomial logistic regression analysis further revealed that the pre-infusion SUVmax value was associated with the risk of developing a higher grade of CRS (*p* < 0.001). These findings suggest that baseline PET metrics, particularly SUVmax, hold promise in predicting CAR T-cell therapy toxicity [25].

#### 4.2.6. General Summary

To summarize, the majority of the studies supported the significance of [^18^F]FDG PET/CT in predicting CAR T-cell therapy toxicity, while only a few disapproved (Table 3).

### 4.3. Diagnostic Utility

In addition to exploring predictive capabilities, few other studies mentioned the valuable role of CAR T-cell therapy in detecting areas involved in CAR T-cell therapy-related toxicity.

#### 4.3.1. First Attempt to Detect Neurotoxicity via Brain [^18^F]FDG PET

The first attempt to investigate this potential was initiated by Rubin et al. in a prospective study [30]. The study enrolled a cohort of 100 patients with a diverse range of hematologic malignancies as well as sarcoma. The most common underlying malignancies included lymphoma (approximately 75%), followed by multiple myeloma, leukemia, and sarcoma. Among these patients, 77 experienced at least one neurological symptom, with encephalopathy being the most frequently reported (57 cases), followed by headache (42 cases), tremor (38 cases), and aphasia (35 cases). The assessment of CAR T-cell toxicity primarily relied on clinical evaluation, with 48 cases exhibiting neurotoxicity graded at a median severity level of 2. The symptoms of neurotoxicity typically began around the sixth day of treatment, with the most severe symptoms occurring on the eighth day. On average, these symptoms lasted for approximately 8.5 days. It was observed that all the patients who experienced neurotoxicity also had CRS. To further investigate the neurotoxicity, various brain imaging techniques were used, including CT, transcranial Doppler ultrasound, magnetic resonance imaging (MRI), CT or MRI angiography, and [^18^F]FDG PET. In six cases where the patients had abnormal electroencephalograms (EEGs), brain [^18^F]FDG PET scans were performed, revealing a focal or diffuse hypometabolism and typically correlating to the findings obtained using functional EEG. The authors also noted evidence of elevated flow velocities observed through transcranial ultrasound in three out of six cases that were assessed using brain [^18^F]FDG PET. Their study demonstrated a strong correlation between the two functional modalities, i.e., brain [^18^F]FDG PET and EEG [30]. These results were disseminated with the primary objective of assisting medical professionals and researchers in identifying, managing, and enhancing the treatment of similar occurrences.

#### 4.3.2. Global Brain Hypometabolism

Global brain hypometabolism is a commonly observed phenomenon in more severe cases of neurotoxicity. A recent report by Aghajan et al. presented a case of spinal myelopathy following CAR T-cell therapy infusion [33]. The patient exhibited an altered mental status, bilateral flaccid paralysis of the lower limbs, and myoclonic jerks in the upper limbs six days after treatment. Functional assessment using EEG revealed generalized periodic discharges, while [^18^F]FDG PET scans showed overall reduced metabolic activity. A brain MRI revealed non-enhancing T2 changes in the external capsule and thalamus on both sides. Additionally, a spinal MRI indicated longitudinal T2 changes, predominantly affecting gray matter throughout most of the spinal cord without enhancement. Following the administration of siltuximab to the patient, there was a notable reduction in the cerebrospinal fluid (CSF) levels of interleukin-6 (IL-6), which correlated with partial clinical improvement. Despite this improvement in the patient’s mental condition, the patient continued to experience paralysis in the lower extremities. A subsequent spinal MRI conducted after 2 months revealed a significant improvement in the abnormality of brain gray matter and a reduction in spinal cord hyperintensity, possibly accompanied by some hemorrhagic changes. However, at the 30-day follow-up, although the patient exhibited normal cognitive function, paralysis in the lower extremities persisted [33].

#### 4.3.3. Frontal Predominant Encephalopathy

Following this, frontal predominant encephalopathy with early paraplegia post-CAR T-cell therapy has been studied more profoundly by Pensato and colleagues [32]. The authors conducted a retrospective study involving four patients diagnosed with DLBCL, with an average age of 66 years and a female predominance of 75%. Prior to the administration of CAR T-cell therapy, a comprehensive neurological screening was performed on these individuals within a timeframe of 15 to 30 days. Following the infusion, they were admitted to the hospital for a minimum of 15 days to closely monitor the onset and progression of neurotoxicity. Subsequently, their progress was followed for a period of up to 180 days post-treatment. The patients in this study were administered either axicabtagene ciloleucel or tisagenlecleucel. All four of the patients experienced CRS within 24 h of CAR T-cell therapy infusion, with an average duration of 19 days. Neurotoxicity was observed in patients 3 and 4 after undergoing chimeric antigen receptor T-cell therapy on days 2 and 3 post-infusion, respectively. In contrast, patients 1 and 2 experienced neurological symptoms simultaneously with a hyperinflammatory state on the day of infusion. Writing disorders were the initial neurological signs observed in all the patients. On average, neurotoxicity manifested 2.3 days after CAR T-cell therapy. Patient 1 had grade 5 ICANS, patient 2 had grade 4 ICANS, patient 3 had grade 3 ICANS, and patient 4 had grade 2 ICANS. Brain MRIs yielded unremarkable results for most of the patients, except for patient 1, who exhibited multiple small T2 hyperintensities in the white matter. EEG recordings revealed generalized slowing in all the patients. Brain [^18^F]FDG PET scans were performed for patients 2, 3, and 4, revealing distinct patterns of hypometabolism. Patient 2 displayed bilateral hypometabolism with frontal predominance (Figure 3); patient 3 showed diffuse hypometabolism with sparing in the posterior region; and patient 4 also exhibited diffuse hypometabolism with sparing in the posterior region.

These findings correlated with the subsequent development of frontal-predominant encephalopathy in all the patients, suggesting that advanced functional imaging techniques such as [^18^F]FDG-PET could serve as valuable tools for assessing ICANS given the absence of distinct diagnostic features [32].

The findings of the preceding two studies were endorsed in a later study. In a subsequent large-scale prospective study by Pensato et al. comprising 46 patients, the median age was 56 years, with 28% being female [29]. Neurotoxicity occurred in 17 of the patients (37%), primarily characterized by encephalopathy that was frequently accompanied by language disturbances (65%) and frontal lobe dysfunction (65%). The median onset time and duration of neurotoxicity were five and eight days, respectively. The brain imaging modalities included [^18^F]FDG PET in 6 of the patients and brain MRIs in all 17 patients with ICANS. Additionally, a neurotoxicity assessment using brain EEG was conducted for all 17 ICANS patients. Among the eight patients evaluated using [^18^F]FDG PET, all displayed hypometabolic patterns. Specifically, five out of six patients exhibited diffuse hypometabolism with frontal predominance, while the remaining patient showed bilateral frontal hypometabolism. Remarkably, functional brain imaging using [^18^F]FDG PET demonstrated complete concordance with the EEG findings in these patients, whereas the brain MRIs were falsely negative in all cases. These findings underscore the utility of [^18^F]FDG PET as a valuable tool for evaluating neurotoxicity in CAR T-cell therapy, particularly given its superior sensitivity compared to conventional brain MRIs [29].

#### 4.3.4. Functional Alterations Associated with Neurotoxicity

Beuchat et al. conducted a retrospective cohort study to review the functional changes associated with neurotoxicity in patients receiving CAR T-cell therapy [40]. The study included 81 patients who received continuous EEG monitoring after CAR T-cell therapy. The median age of the patients was 60 years, with a female representation of 35.8%. All the patients experienced CRS, with neurotoxicity typically occurring approximately 6 days after CAR T-cell therapy. The severity of neurotoxicity varied, with grade 3 neurotoxicity affecting the majority of the patients (51.9%). Brain [^18^F]FDG PET imaging revealed various patterns of hypometabolism in eight of the patients, with some patients showing diffuse hypometabolism and others displaying lateralized or hemispheric hypometabolism. Only one patient was identified with normal [^18^F]FDG PET imaging results. The EEG findings were consistent with the imaging results, demonstrating rhythmic and periodic patterns in seven out of eight of the patients. Notably, brain [^18^F]FDG PET imaging provided more accurate insights than CT or MRI [40].

#### 4.3.5. Irreversible Neurotoxicity

Gust and colleagues explored the potential of brain [^18^F]FDG PET imaging for detecting irreversible neurotoxicity [34]. This prospective cohort study enrolled 43 B-ALL patients with a median age of 12 years. Among these patients, 19 exhibited signs of neurotoxicity, with 10 experiencing mild to moderate symptoms and 9 manifesting more severe presentations. The onset of neurological symptoms typically occurred approximately 7 days post-therapy, reaching peak severity approximately 1 day later, and persisted for a median duration of 5 days. While the study primarily focused on clinical and biochemical profiles, the authors did report an intriguing molecular imaging finding in one patient. Initially, a brain MRI was performed after the patient exhibited clinically evident neurotoxicity. The MRI revealed diffusion restrictions in both occipital gyri, suggesting the presence of cortical cytotoxic edema. This edema resolved on a subsequent MRI scan conducted five weeks later. However, the patient developed epilepsy, prompting brain [^18^F]FDG PET imaging. The scan revealed volume loss and reduced [^18^F]FDG uptake in the right occipital cortex, indicating irreversible neuronal injury induced by the CAR T-cell therapy [34]. These findings highlight the superiority of [^18^F]FDG PET over MRI in detecting and assessing the neuronal consequences of this therapy.

#### 4.3.6. Tracking CAR T-Cell Therapy Toxicity Elsewhere

As previously mentioned, the assessment of postoperative CAR T-cell therapy can be improved by incorporating follow-up PET/CT scans at specific time intervals (M1 and M3). This approach offers the advantage of ensuring a comprehensive evaluation of the entire body. Some researchers have reported a few instances of CAR T-cell toxicity outside of the central nervous system during these time points. For example, a case of pneumonia associated with CAR T-cell therapy was documented in a review by Huang et al. [42]. During the M1 PET/CT scan, a 47-year-old female patient who had undergone CAR T-cell therapy displayed hypermetabolic diffuse bilateral basal lung parenchymal ground glass densities four weeks after treatment [42]. In addition, in their predictive study mentioned earlier, Wang et al. also discussed the detection of CAR T-cell therapy-induced myocarditis using [^18^F]FDG PET/CT in a single patient [31]. The patient in question was a 35-year-old male with DLBCL affecting various body parts including the myocardium. Following the CAR T-cell infusion, the patient experienced worsening shortness of breath and was found to have pericardial effusion and elevated troponin levels. To rule out pseudoprogression, a follow-up [^18^F]FDG PET/CT scan was conducted after 2 weeks, which showed partial disease control but evidence of increased [^18^F]FDG uptake in the myocardium, indicating myocarditis. It is worth noting that subsequent [^18^F]FDG PET/CT scans revealed an improvement in CAR T-cell therapy-induced myocarditis, with decreased myocardial FDG uptake and improved pericardial effusion [31].

## 5. Discussion

### 5.1. Significance of the Current Review

This study provides the first systematic review of PET/CT utilities in CAR T-cell therapy toxicity, a field of significant concern in cancer immunotherapy. Our findings indicate that [^18^F]FDG PET imaging offers a wide range of capabilities and demonstrates reliability in the majority of studies. Overall, PET imaging is a sensitive and dependable tool that is essential for predicting, detecting, and monitoring CAR T-cell therapy neurotoxicity. Furthermore, in cases of neurotoxicity, there may be a need for immediate posttherapy PET/CT evaluation on a case-by-case basis to supplement EEG assessments, as MRI or CT scans alone may not be sufficient.

### 5.2. Advancement of Preclinical Research

#### 5.2.1. [^18^F]-Labeled Tracers

In addition to the previously discussed valuable insights gained by utilizing PET/CT imaging with [^18^F]FDG used as radiotracer, molecular imaging assessment extends beyond [^18^F]FDG, as preclinical studies have concluded.

[^18^F]tetrafluoroborate ([^18^F]TFB) PET is an advanced molecular imaging modality that utilizes the radiotracer [^18^F]TFB, a fluorine-18 labeled compound. This tracer is particularly effective for imaging the sodium iodide symporter (NIS), which is an intrinsic cellular mechanism used by thyroid cells to concentrate iodide. The utility of [^18^F]TFB in PET imaging stems from its ability to be taken up by cells expressing NIS, thereby enabling the visualization of biological processes that involve NIS-expressing cells. The imaging mechanism of [^18^F]TFB PET involves the administration of the radiotracer, which circulates and is selectively absorbed by cells that express the NIS [43].

[^18^F]TFB PET has been applied preclinically to track and monitor CAR T-cell therapy. In these therapies, CAR T cells are engineered to express NIS, allowing [^18^F]TFB PET to track the distribution, expansion, and persistence of these cells in vivo. This capability is crucial for assessing the efficacy and safety of CAR T-cell therapies, providing insights into therapeutic outcomes and potential toxicities such as CRS.

Sakemura et al. initiated the first preclinical study to determine in vivo pharmacokinetics [43]. To address this, the authors utilized the NIS as a platform to image and track CAR T cells. They engineered CD19-directed and B-cell maturation antigen-directed CAR T cells to express NIS (NIS^+^CART19 and NIS^+^BCMA-CART, respectively) and tested the sensitivity of [^18^F]tetrafluoroborate ([^18^F]TFB) to detect trafficking and expansion in systemic and localized mouse tumor models and in a CAR T-cell therapy toxicity model. [^18^F]TFB PET was used to detect CAR T-cell trafficking to tumor sites and in vivo expansion, which was correlated with the development of clinical and laboratory markers of CRS. These studies demonstrate a noninvasive, clinically relevant method to assess CAR T-cell functions in vivo. In mice that experienced symptoms of CRS following treatment with NIS^+^CART19 cells, the use of [^18^F]TFB PET imaging revealed significant uptake of [^18^F]TFB in various organs including the bone marrow, spleen, liver, and lungs. Conversely, in mice treated with NIS^+^CART19 cells who did not develop CRS symptoms, [^18^F]TFB PET imaging showed normal levels of [^18^F]TFB uptake. An analysis of the quantitative data demonstrated that the mice with CRS had significantly higher levels of [^18^F]TFB uptake compared to those without CRS. This study provides evidence of a potential correlation between [^18^F]TFB uptake, the development of CRS symptoms, T-cell expansion, and elevated cytokine levels in this model of CAR T-cell therapy [43].

#### 5.2.2. Non-[^18^F]-Labeled Tracers

Beyond [^18^F]-labeled compounds, the optimization of [^89^Zr]Zr-oxine labeling of CAR T cells has been meticulously refined to ensure that subsequent in vivo PET imaging faithfully reflects the localization and persistence of these cells post-infusion [44]. This optimization encompasses the determination of the maximum [^89^Zr]Zr-oxine labeling-specific activity that preserves CAR T-cell viability and functionality, thereby supporting the successful deployment of CAR T-cell therapies [44]. The utilization of [^89^Zr]Zr-oxine-labeled tracers facilitates noninvasive visualization and quantification of CAR T cells in vivo [45]. This capability assumes paramount significance in evaluating the initial distribution of CAR T cells, their trafficking to tumor loci, and their enduring presence over time, all of which are pivotal determinants of therapeutic efficacy. Evidence from studies corroborates that [^89^Zr]Zr-oxine labeling has no deleterious impact on the biological functions of CAR T cells, including their viability, proliferation, cytotoxicity, and cytokine productivity [44,45]. Furthermore, [^89^Zr]Zr-labeled tracers may have the potential to be used to evaluate the safety profile and potential toxicity of CAR T-cell therapies [45]. Through meticulous tracking of CAR T-cell distribution, researchers can discern any inadvertent accumulation in nontarget tissues and prospectively provide countermeasures before clinical manifestations [44,45]. Beyond their clinical application, [^89^Zr]Zr-labeled tracers will serve as valuable tools for the research and development of novel CAR T-cell therapies in the near future.

#### 5.2.3. Interleukin-Focused Imaging Approach

Recently, researchers have focused on incorporating interleukins, which play a significant role in the development of CRS and ICANS, into the immunoPET concept at the preclinical level. Leland et al. conducted a study to determine whether radiolabeling of CAR T cells could facilitate biodistribution studies using PET [45]. They labeled CAR T cells that targeted the interleukin (IL-13Rα2) receptor with [^89^Zr]Zr-oxine and then compared their characteristics with those of non-labeled CAR T cells. The researchers optimized the conditions for labeling with [^89^Zr]Zr-oxine, including the incubation time, temperature, and the use of serum. They also assessed the T-cell subtypes and various attributes of the radiolabeled CAR T cells to evaluate their overall quality, including cell viability, proliferation, activation and exhaustion markers, cytolytic activity, and release of interferon-γ when co-cultured with glioma cells expressing IL-13Rα2. The researchers observed that the process of radiolabeling CAR T cells with [^89^Zr]Zr-oxine is rapid and effective, with the radioactivity remaining in the cells for at least 8 days with minimal loss. Additionally, there were no significant differences in the expression of T-cell activation markers or T-cell exhaustion markers between the radiolabeled and unlabeled CAR T cells. It is worth noting that radiolabeling has negligible effects on the biological properties of CAR T cells, specifically in terms of their effectiveness in targeting IL-13Rα2-positive tumor cells [45]. Therefore, CAR T cells targeted at IL-13Rα2 and radiolabeled with 89Zr-oxine maintain essential product attributes, suggesting that radiolabeling with [^89^Zr]Zr-oxine could aid in studying biodistribution and tissue trafficking in vivo using PET imaging.

### 5.3. General Perspectives

Up until now, PET/CT and molecular imaging have not been fully recognized for their value in areas beyond assessing therapeutic responses. This lack of recognition became apparent during our research screening process. The majority of studies and clinical trials, as well as researchers, have primarily focused on using PET/CT for standard response evaluation and predicting disease survival. As a result, there has been little interest in exploring its predictive or diagnostic capabilities for CAR T-cell therapy. We also encountered review articles that extensively discussed CAR T-cell toxicity but did not acknowledge PET/CT’s predictive and diagnostic potential [46,47]. By presenting this substantial evidence, we hope to encourage greater recognition of these underestimated values in the future, especially considering the growing interest and availability of CAR T cells worldwide.

### 5.4. Authors’ Thoughts on Mechanisms and Improvements

The integration of immunoPET and artificial intelligence (AI) may represent a transformative approach to enhancing the precision of in vivo tracking imaging systems. ImmunoPET, which combines the specificity of antibodies with the sensitivity of PET imaging, offers a robust platform for visualizing the dynamic processes of immune cells within the body [14]. The application of AI further augments this capability by enabling the analysis of complex imaging data, potentially identifying patterns and correlations that may escape human detection [12]. Together, these technologies could significantly refine the assessment of therapy response and its associated side effects. This is crucial for monitoring the immediate effects of therapy and for predicting long-term outcomes. AI can contribute to this field by facilitating the interpretation of imaging data, leading to more accurate predictions of therapeutic efficacy and toxicity. For instance, machine learning algorithms can be trained to recognize early signs of CRS or neurotoxicity, potentially enabling preemptive interventions. The advancement of these technologies is not without challenges. The development of novel immunoPET tracers and the refinement of AI algorithms require extensive validation in clinical settings. Moreover, the integration of AI into clinical workflows must be handled with care to ensure that the technology supports, rather than supplants, the clinical judgment of healthcare professionals.

### 5.5. Study Limiations

It is important to note that this article has certain limitations, including its qualitative approach, small sample size, and inclusion of diverse cancer subtypes. Nonetheless, it is the only study to date that investigates the promising domains revealed by PET/CT reliance in the context of CAR T-cell therapy toxicity.

## 6. Conclusions

[^18^F]FDG PET/CT, beyond its role in assessing therapy response and prognostication, provides added advantages for CAR T-cell therapy recipients. In the pre-therapeutic landscape, baseline PET-derived metrics have demonstrated predictive insights and correlated with CAR T-cell therapy toxicity incidence, as indicated by the majority of the reviewed research. Moreover, post-therapeutic PET/CT at the M1 or M3 timepoints can be used to diagnose and monitor CAR T-cell therapy-induced toxicity with great reliability, especially for brain neurotoxicity, through functional insights. Such evidence should stimulate further research endeavors in prospective studies and clinical trials to establish generalizability and acknowledgment.

## Figures and Tables

**Figure 1 cancers-16-01728-f001:**
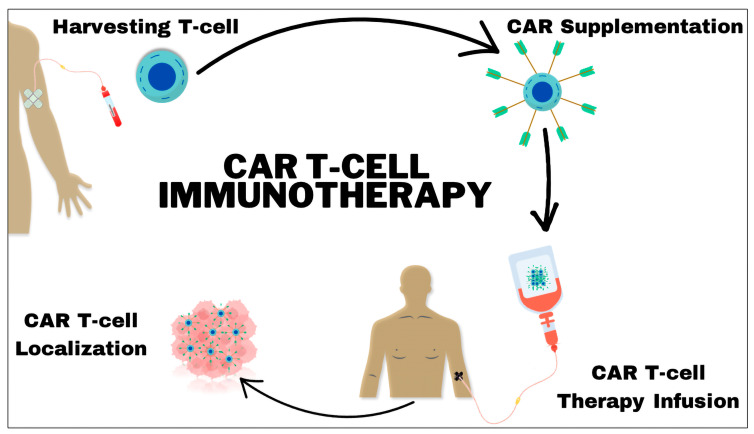
Illustration of the foundational concept of chimeric antigen receptor (CAR) T-cell therapy. T cells are extracted from the patient and genetically engineered in a laboratory setting by incorporating CARs. Following this modification process, the enhanced T cells are reintroduced into the patient. These engineered cells stimulate the host immune system to target tumor cells through the recognition of and direct binding to particular tumor antigens.

**Figure 2 cancers-16-01728-f002:**
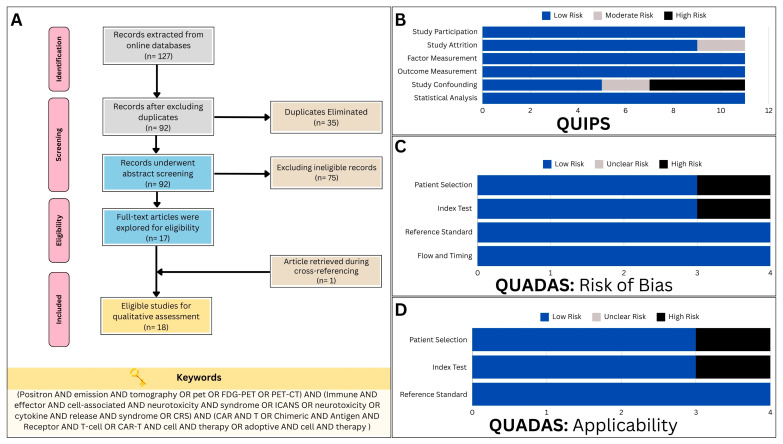
(**A**) The study flow diagram, constructed in accordance with the guidelines outlined in the Preferred Reporting Items for Systematic Review and Meta-Analysis Protocols (PRISMA-P) statement, illustrates the process of study selection and inclusion. (**B**) Assessment of bias risk using the Quality in Prognostic Studies (QUIPS) tool for predictive studies incorporated in the systematic review. (**C**) Assessment of bias risk using the Quality Assessment of Diagnostic Accuracy Studies (QUADAS-2) protocol for diagnostic studies incorporated in the systematic review. (**D**) Assessment of applicability concerns using the Quality Assessment of Diagnostic Accuracy Studies (QUADAS-2) protocol for the diagnostic studies incorporated in this systematic review.

**Figure 3 cancers-16-01728-f003:**
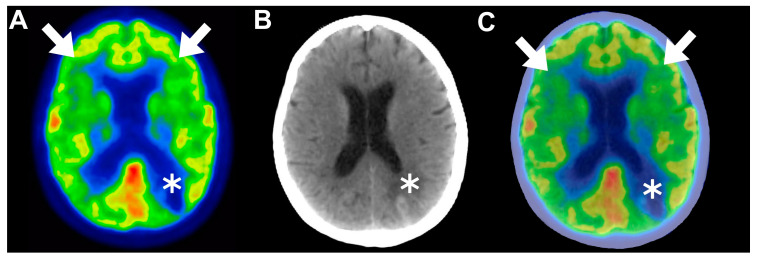
(**A**–**C**) Matched axial views of brain 2-deoxy-2-[18F]fluoro-D-glucose positron emission tomography ([^18^F]FDG PET), computed tomography (CT), and fused PET/CT images in a 65-year-old female patient with grade 4 immune effector cell-associated neurotoxicity syndrome (ICANS) neurological manifestations developed approximately 40 days after CAR T-cell therapy infusion. Functional brain images revealed diffuse hypometabolism showcasing bifrontal predominance (arrows); in addition, a single hypometabolic hyperdensity was observed in the left parietal lobe and was correlated with cerebral hematoma (asterisk). This representative case demonstrates the superiority of [^18^F]FDG PET in detecting ICANS by providing functional insights that are undeliverable using the conventional CT modality. This figure was initially presented in a publication by Pensato et al. [32] and was authorized for use under a Creative Commons Attribution 4.0 International License. Panels A and D of Figure 4 [32] were recovered and enhanced with annotations using manual image fusion techniques facilitated by MATLAB software (Version R2023b).

**Table 1 cancers-16-01728-t001:** The assessment of methodological quality in studies focused on the use of PET for evaluating toxicity related to chimeric antigen receptor (CAR) T-cell therapy in a singular context.

CARE Item ^1^	Aghajan 2021 [33]	Gust 2019 [34]	Huang 2023 [42]
Title	0	0	0
Keywords	0	1	1
Abstract	1	1	1
Introduction	1	1	1
Patient information	1	1	1
Clinical findings	1	1	1
Timeline	0	1	0
Diagnostic assessment	1	1	1
Therapeutic intervention	1	1	0
Follow-up and outcome	1	0	0
Discussion	0	0	0
Patient perspective	0	0	0
Informed consent	1	1	0
Numerical score	8	9	6
Quality score	Intermediate quality	Intermediate quality	Intermediate quality

^1^ CARE: Case report guidelines.

**Table 2 cancers-16-01728-t002:** Characteristics of the included studies in the systematic review.

First Author (Year, Country)	Patients (M ^1^, F ^2^)	Age in Years	Cancer Type	Article Type (Design)	Institutional Experience	Patients Evaluated via PET	Explored Domain
Ababneh (2023, USA ^3^) [39]	59 (33 M, 26 F)	66	LBCL ^4^ (r/r ^5^)	Original (R ^6^)	Monocentric	59 Patients	Predictive
Beuchat (2022, USA) [40]	91 (52 M, 29F)	61	NHL ^7^	Original (R)	Monocentric	8 Patients	Diagnostic
Derlin (2021, DE ^8^) [41]	10 (6 M, 4 F)	59	DLBCL ^9^ (r/r)	Original (R)	Monocentric	10 Patients	Predictive
Gui (2024, CN ^10^) [25]	38 (23M, 15 F)	55	DLBCL ^9^ (r/r)	Original (R)	Monocentric	38 Patients	Predictive
Hong (2021, CN) [26]	41 (17 M, 24 F)	44	NHL (r/r)	Original (R)	Monocentric	44 Patients	Predictive
Marchal (2024, FR ^11^) [27]	56 (36 M, 20 F)	60	LBCL	Original (R)	Multicentric	56 Patients	Predictive
Morbelli (2023, IT ^12^) [28]	21 (10 M, 11 F)	56	DLBCL (rf ^13^)	Original (R)	Monocentric	21 Patients	Predictive
Pensato (2023, IT) [29]	46 (33 M, 13 F)	56	NHL	Original (P ^14^)	Monocentric	6 Patients	Diagnostic
Rubin (2019, USA) [30]	100 (61 M, 39 F)	64	HM ^15^ and Sarcoma	Original (P)	Monocentric	6 Patients	Diagnostic
Wang (2019, CN) [31]	19 (12 M, 7 F)	43	NHL	Original (R)	Monocentric	19 Patients	Predictive
Pensato (2022, IT) [32]	4 (1 M, 3 F)	66	DLBCL	Original (R)	Monocentric	4 Patients	Diagnostic
Gust (2019, USA) [34]	43 (21 M, 22 F)	12	B-ALL ^16^	Original (P)	Monocentric	1 Patient	Diagnostic
Cohen (2021, IL ^17^) [35]	48 (25 M, 23F)	68	DLBCL (r/r)	Original (R)	Monocentric	48 Patients	Predictive
Iacoboni (2021, ES ^18^) [36]	35 (26 M, 9F)	58	LBCL (r/r)	Original (R)	Monocentric	35 Patients	Predictive
Ligero (2023, ES) [37]	93 (63 M, 30 F)	59	LBCL (r/r)	Original (R)	Monocentric	93 Patients	Predictive
Voltin (2022, DE) [38]	47 (29 M, 18 F)	61	LBCL (r/r)	Original (R)	Multicentric	47 Patients	Predictive
Aghajan (2021, USA) [33]	1 M	30	PMBCL ^19^	Case report	NA ^20^	1 Patient	Diagnostic
Huang (2023, USA) [42]	1 F	47	B-ALL	Review article	NA	1 Patient	Diagnostic

^1^ M: Male; ^2^ F: female; ^3^ USA: United States of America; ^4^ LBCL: large B-cell lymphoma; ^5^ r/r: relapsed/refractory; ^6^ R: retrospective; ^7^ NHL: non-Hodgkin’s lymphoma; ^8^ DE: Germany; ^9^ DLBCL: diffuse large B-cell lymphoma; ^10^ CN: China; ^11^ FR: France; ^12^ IT: Italy; ^13^ rf: refractory; ^14^ P: prospective; ^15^ HM: hematologic malignancy; ^16^ B-ALL: B-cell precursor acute lymphoblastic leukemia; ^17^ IL: Israel; ^18^ ES: Spain; ^19^ PMBCL: primary mediastinal B-cell lymphoma; ^20^ NA: not applicable.

**Table 3 cancers-16-01728-t003:** Summary table of predictive research results.

Studies Reporting Unpredictability		Studies Reporting Predictability	
First Author, Year	Examined PET Parameters	First Author, Year	Examined PET Parameters
Iacoboni, 2021 [36]	Baseline (SUVmax ^1^, TMTV ^2^)	Wang, 2019 [31]	Baseline (SUVmax, TMTV *, TLG *)
Cohen, 2021 [35]	Baseline (SUVmax, TMTV, TLG ^3^)	Derlin, 2021 [41]	Baseline (SUVmax *, TMTV, TLG)
Voltin, 2022 [38]	Baseline (SUVmax, TMTV)	Hong, 2021 [26]	Baseline (SUVmean ^4^*, TMTV *, TLG *)
Ligero, 2023 [37]	AI ^5^ radiomics	Marchal, 2024 [27]	Baseline (SUVmean; liver * and spleen *)
		Morbelli, 2023 [28]	Baseline (TMTV *, TLG *)
		Ababneh, 2023 [39]	Baseline (SUVmax *, TMTV *, TLG *)
		Gui, 2024 [25]	Baseline (SUVmax *, TMTV, TLG *)

^1^ SUVmax: Maximum standardized uptake value; ^2^ TMTV: total metabolic tumor volume; ^3^ TLG: total lesion glycolysis; ^4^ SUVmean: mean standardized uptake value; ^5^ AI: artificial intelligence; *: statistically significant parameter.

## Data Availability

The data presented in this study are available on request from the corresponding author. The data are not publicly available due to privacy.

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
