# Peer review of "FDG-PET in Chimeric Antigen Receptor T-Cell (CAR T-Cell) Therapy Toxicity: A Systematic Review"

_cancers, 2024, doi:10.3390/cancers16091728_

Round 1
Reviewer 1 Report
Comments and Suggestions for Authors
The author Al-Ibraheem et. al in their ms titled “PET in Chimeric Antigen Receptor T-cell Therapy Toxicity: A Systematic Review” is specifically tailored to examine the utility of PET in evaluating adverse events induced by CAR T-cell therapy. Over all the manuscript is well supported by data driven information. However, I have few concern before acceptation.
Minor Comment:
1. The basics of CART cell like what and how this generated is missing in the introduction section.
Author Response
Dear Review 1,
We truly appreciated your respectful feedback and would like to express our sincere gratitude for providing the time and effort to review this research work.
In response to your respectful review point, we have modified the introduction section to discuss the basics of CAR T-cell therapy as suggested. Kindly track changes highlighted in yellow color (lines 65-82).
Reviewer 2 Report
Comments and Suggestions for Authors
PET in Chimeric Antigen Receptor T-cell Therapy Toxicity: A Systematic Review
This is an extensive review of CAR T-cell therapy's toxicity to cancer and its side effects using PET imaging techniques. It’s interesting and summarizes FDG PET very well. This reviewer has some concerns.
1. The review is not well organized: consider adding subtitles to avoid paragraphs that are too long and hard to follow.
2. The [18F]FTB PET should have a definition and a short description of the imaging mechanisms and successful applications for disease assessments.
3. The authors missed other PET tracers related to CAR T-cell therapy prognosis, such as PSMA-based PET/CT radiotracer [18F]DCFPyL and [18F]-TMP, et al.
4. [89Zr] and FTB PET should be reviewed better, and PET tracers should be discussed in Section 3. And include a summary table of all tracers related to CAR T-cell therapy PET imaging.
5. The author may discuss perspectives of more specific PET imaging strategies for assessing toxicity to tumors, brains, or other target organs in the discussion section.
6. The review is more related to literature and lacks the authors’ thoughts on mechanisms and improvements.
Comments on the Quality of English LanguageThe language is okay and no major concerns. Some minor typos and grammatical issues.
Long Paragraphs should be restructured.
Author Response
Dear Reviewer 2,
We sincerely appreciate your valuable input. We have addressed each of your esteemed comments as provided below.
- The review is not well organized: consider adding subtitles to avoid paragraphs that are too long and hard to follow.
- We have re-organized the text using subheadings.
- We have re-organized the text using subheadings.
- The [18F]FTB PET should have a definition and a short description of the imaging mechanisms and successful applications for disease assessments.
- In response to your respectful review point, we have added short definition and description for [18F]FTB PET. Kindly track changes in lines 491-506 (Highlighted in yellow).
- In response to your respectful review point, we have added short definition and description for [18F]FTB PET. Kindly track changes in lines 491-506 (Highlighted in yellow).
- The authors missed other PET tracers related to CAR T-cell therapy prognosis, such as PSMA-based PET/CT radiotracer [18F]DCFPyL and [18F]-TMP, et al.
- Thank you for bringing these studies to our attention. We appreciate your interest in our systematic review. Please note that our review was specifically designed to focus on clinical (in-human) studies concerning CAR T-cell toxicity rather than prognosis. As indicated in lines 137-138 of our manuscript, we included only articles that "specifically addressed the application of PET in the context of CAR T-cell therapy toxicity in real clinical settings." Consequently, these particular radiotracers fall outside the scope of our review and could not be included. We hope this clarification addresses your concerns.
- Thank you for bringing these studies to our attention. We appreciate your interest in our systematic review. Please note that our review was specifically designed to focus on clinical (in-human) studies concerning CAR T-cell toxicity rather than prognosis. As indicated in lines 137-138 of our manuscript, we included only articles that "specifically addressed the application of PET in the context of CAR T-cell therapy toxicity in real clinical settings." Consequently, these particular radiotracers fall outside the scope of our review and could not be included. We hope this clarification addresses your concerns.
- [89Zr] and FTB PET should be reviewed better, and PET tracers should be discussed in Section 3. And include a summary table of all tracers related to CAR T-cell therapy PET imaging.
- Thank you for your suggestion. However, incorporating these radiotracers would contradict the methodology outlined by the PRISMA-P statement for systematic reviews. We included brief mentions of these radiotracers to provide insights from promising preclinical studies focused on in-vivo tracking. Our systematic review primarily focuses on clinical studies utilizing FDG as the primary radiotracer. For your kind reference, a summary table of the included studies can be found on pages 6-7 of our manuscript. We hope this explanation clarifies our approach.
- Thank you for your suggestion. However, incorporating these radiotracers would contradict the methodology outlined by the PRISMA-P statement for systematic reviews. We included brief mentions of these radiotracers to provide insights from promising preclinical studies focused on in-vivo tracking. Our systematic review primarily focuses on clinical studies utilizing FDG as the primary radiotracer. For your kind reference, a summary table of the included studies can be found on pages 6-7 of our manuscript. We hope this explanation clarifies our approach.
- The author may discuss perspectives of more specific PET imaging strategies for assessing toxicity to tumors, brains, or other target organs in the discussion section.
- We have discussed a more specific interleukin-directed immunoPET approach according to your respectful review point. Kindly track changes in lines 544-565 (Highlighted in yellow).
- We have discussed a more specific interleukin-directed immunoPET approach according to your respectful review point. Kindly track changes in lines 544-565 (Highlighted in yellow).
- The review is more related to literature and lacks the authors’ thoughts on mechanisms and improvements.
- We have added a new subheading to discuss authors’ thoughts on mechanisms and improvements. Kindly track changes in lines 578-596 (Highlighted in yellow).

Round 2
Reviewer 2 Report
Comments and Suggestions for Authors
Thanks for the revision to address my concern from the first round of discussions.
1. Please change the title to reflect the specificity:
FDG-PET in Clinical Chimeric Antigen Receptor T-cell Therapy Toxicity: A Systematic Review
2. Please reorganize and consider using subheadings for 3.4 and 3.5.1.
Comments on the Quality of English Language
Minor typos.
Author Response
Dear Reviewer 2,
Thank you once again for taking the time and effort to reinspect this manuscript and provide further comments to strengthen the point of view shared in this article.
Below are point-by-point response to your respectful review points
- Please change the title to reflect the specificity: “FDG-PET in Clinical Chimeric Antigen Receptor T-cell Therapy Toxicity: A Systematic Review”
- Your suggestion is much appreciated. We have implemented your suggested title.
- Your suggestion is much appreciated. We have implemented your suggested title.
- Please reorganize and consider using subheadings for 3.4 and 3.5.1.
- In response to your respectful review, reorganization has been made and is highlighted in yellow.
